# Reviewing Chitin/Chitosan Nanofibers and Associated Nanocomposites and Their Attained Medical Milestones

**DOI:** 10.3390/polym13142330

**Published:** 2021-07-16

**Authors:** Iyyakkannu Sivanesan, Judy Gopal, Manikandan Muthu, Juhyun Shin, Jae-Wook Oh

**Affiliations:** 1Department of Bioresources and Food Science, Konkuk University, Seoul 143-701, Korea; isivanesan@gmail.com; 2Laboratory of Neo Natural Farming, Chunnampet 603 401, Tamil Nadu, India; jejudy777@gmail.com (J.G.); bhagatmani@gmail.com (M.M.); 3Department of Stem Cell and Regenerative Biotechnology, Konkuk University, Seoul 143-701, Korea; junejhs@konkuk.ac.kr

**Keywords:** chitosan nanoparticles, chitin, nanofibers, nanocomposites, synthesis, medical applications

## Abstract

Chitin/chitosan research is an expanding field with wide scope within polymer research. This topic is highly inviting as chitin/chitosan’s are natural biopolymers that can be recovered from food waste and hold high potentials for medical applications. This review gives a brief overview of the chitin/chitosan based nanomaterials, their preparation methods and their biomedical applications. Chitin nanofibers and Chitosan nanofibers have been reviewed, their fabrication methods presented and their biomedical applications summarized. The chitin/chitosan based nanocomposites have also been discussed. Chitin and chitosan nanofibers and their binary and ternary composites are represented by scattered superficial reports. Delving deep into synergistic approaches, bringing up novel chitin/chitosan nanocomposites, could help diligently deliver medical expectations. This review highlights such lacunae and further lapses in chitin related inputs towards medical applications. The grey areas and future outlook for aligning chitin/chitosan nanofiber research are outlined as research directions for the future.

## 1. Introduction

Miniaturization has become the recent state-of-the-art technology, resulting in modifying material shapes, surfaces, characteristics, and functions [1]. Nanotechnology leads the forefront in the arena of miniaturization strategies. The impressive potential applications of nanotechnology have transformed this field and opened it for many researchers and scientists to work on [2]. Nanotechnology today has expanded into different scientific areas, extending from electronics to industrials to cosmetics [3,4,5]. Nanotechnology has also strongly impacted core areas of human health and welfare.

Polymers of natural origins have been extensively employed not only in the food industry but also in pharmaceuticals [6]. Chitin, chitosan, alginate, and carrageenan are the most commonly used polysaccharide polymers in various pharmaceutical applications [7,8,9,10,11,12,13] and food quality, safety, and preservation. Polysaccharide polymers are sought after owing to their non-toxicity, biocompatibility, and biodegradability [7,8]. Chitin (Ch) consists of repeated N acetyl-D-glucosamine (GlcNAc) units linked by β-(1,4) glycosidic bonds [14]. Naturally, chitin exists in three crystalline polymorphic forms. Within each form, there are different orientations of the microfibrils: α-chitin has parallel chains, β-chitin has antiparallel chains, and γ-chitin has a mixture of parallel and anti-parallel chains (Figure 1) [15]. Figure 1 displays the representative organisms for each of these polymorphs.

Chitin is insoluble in water; however, water-soluble chitosan (ChS) has been produced after a chitin deacetylation process. A ChS molecule contains an –NH_2_ group and two –OH radicals in each glycoside residue. Although chitin is naturally produced at a rate of around 1010 tons per year, most chitin is thrown away as commercial waste [15]. Chitin is disposed of as marine shell wastes or industrial wastes in the magnitude of multiple tons. ChS, a polymer of interest, is used to deliver different therapeutic agents, increasingly known for its use in pharmaceuticals. This is attributed to its biocompatibility, ability to bind few organic compounds, susceptibility to enzymatic hydrolysis, and intrinsic physiological activity combined with nontoxicity [16,17,18]. These properties make ChS amenable to a wide variety of biomedical applications, including drug delivery and targeting, wound healing, tissue engineering, and similar allied fields. Moreover, ChS is also known for its antimicrobial properties, as well as color stabilization, emulsification, antioxidant activities, and dietary fiber-like properties, water-retention, and fat entrapment. Chitosan has attracted great attention as a material for drug delivery biomedical applications as a promising material specifically for delivering macromolecules [19,20,21]. Therapeutic agents have been incorporated into such polymeric matrix to protect the biologically active compound from degradation, promote controlled drug release and improve absorption and therapeutic effect. Miniaturization of chitosan has also been attempted, with ChS-based nanomaterials ranging from microparticles to NP composites and nanofilms, showing enhanced biomedical properties.

A nanofiber (NF) is generally defined as a fiber less than 100 nm diameter and an aspect ratio greater than 100:1 [22,23]. Properties of NFs are distinct from that of microfibers, because NFs have a characteristic morphology, an extremely high surface-to-volume ratio [24], and optical [25] and mechanical properties. In addition to these, they are biodegradable, biocompatible, renewable, and sustainable [26]. The electro-spinning process is well known for producing artificial NFs from a wide range of polymers [27,28]. Owing to its linear structure, chitin has high crystallinity and is arranged as NFs; these NFs are embedded in a protein matrix. Crab and prawn shells also have a complex hierarchical structure consisting of NFs [29].

In this current review, we briefly run through the existing Ch and ChS based nanomaterial synthesis and their applications. The biomedical applications of Ch and ChS nanofibers have been dealt with elaborately in this review. The future perspective of Ch and ChS NFs is discussed and the reason why Ch NFs are less applied for biomedical applications than ChS NFs speculated.

## 2. Chitin and Chitosan Nanomaterials

### 2.1. Overview of Ch/ChS Nanomaterial Preparation Methods

Ch is recovered from natural sources by a two-step process. The first step is the extraction of Ch and removal of calcium carbonate (CaCO_3_) from crustacean shells using dilute hydrochloric acid, followed by deproteination using dilute aqueous NaOH. During the second step, 40–50% aqueous NaOH at 110–115 °C is used for the deacetylation of Ch. When the degree of deacetylation exceeds 50%, then ChS is produced [30]. Ch with a degree of deacetylation of ≥75% is also recognized as ChS [31]. Elgadir et al., 2015 have elaborately reviewed these preparation methods [32].

ChS NPs were first characterized by Ohya et al. (1994) for the delivery of 5-fluorouracil, a chemotherapy medication [33]. The basic techniques demonstrated for the preparation of ChS NPs revolve around emulsification, precipitation, ionic or covalent crosslinking, or combinations thereof. Emulsification and crosslinking are the pioneering methods demonstrated in literature that are based on the amino group of ChS and the aldehyde group of a crosslinking agent [33]. The reversed micelles (microemulsion) method based on covalent crosslinking, has also been applied for the effective preparation of ChS NPs. ChS NPs can also be produced by precipitation-based methods. The phase inversion precipitation method combines emulsification and precipitation techniques. Alternatively, the emulsion-droplet coalescence method, also called desolvation, has been described, based on the coalescence of two water-in-oil emulsions which induce precipitation of ChS NPs. Ionic gelation was first described by Calvo et al. (1997) [34]; this method can also be used in combination with radical polymerization, which induces gelation of chitosan simultaneously with polymerization of acrylic or methacrylic acid [35]. ChS NPs can also be produced via self-assembly, which is a widely used method based on multiple electrostatic, or hydrophobic and/or van der Waals force based interactions that may be between ChS and other molecules [36,37].

Recently, green preparation routes and the use of “mild” compounds have become popular. Spray drying is one such method; ChS in aqueous acetic acid is passed through a nozzle exposed to air temperatures from 120–150 °C, resulting in ChS NPs [38]. Magnetic ChS NPs have been produced using this method [39]. Supercritical-CO_2_- assisted solubilization and atomization (SCASA) is one of the pioneering green methods as it is a process that is free of acid- or harmful solvent and uses only water and CO_2_. The dissolution of ChS in water occurs through the acidifying effect of pressurized CO_2_ under high pressure. After a 48 h dissolution step, the ChS solution is fed in for atomization. NPs form due to the drying process and are collected by a filter [40,41]. The long processing time and the size of the particles and their distribution are the disadvantages of this method. There are few other green preparation methods reported for ChS NP synthesis, based on ultrasonication, microwave synthesis and microbe-based methods [42].

There are almost no reports on the preparation of Ch NP, except for the report by Lu et al. [43], who demonstrated a route for preparing suspensions of Ch crystallite particles. However, Ch nanowhiskers have been reported by several authors. Ch was treated with strong HCl, followed by controlled hydrolysis to obtain Ch nanowhiskers. Another method used for Ch nanowhiskers’ isolation is TEMPO mediated oxidation (2,2,6,6-tetramethylpiperidine-1-oxyl radical mediated oxidation). Whiskers have recently been prepared from Ch from different sources, such as crab, shrimp, and lobster shells and squid pens [44,45]. Morin and Dufresne [46] prepared nanochitin whiskers from Riftia (marine invertebrates). In another study, Gopalan and Dufresne obtained nanochitin whiskers from crab shells. Rujiravanit et al. reported the preparation of Ch whiskers via acid hydrolysis of shrimp shells [47]. Revol et al. reported the preparation of a suspension of chitin crystallites of rod-like particles through acid hydrolysis [48,49], with average sizes of 200 ± 20 nm in length and 8 ± 1 nm in width. Salaberria et al. extracted chitin from yellow lobster shell wastes followed by the isolation of nanocrystals by acid hydrolysis [49]. Zeng et al. [44] and Morin et al. [45] have covered more details on this topic in their reviews.

### 2.2. Biomedical Applications of Ch/ChS Nanomaterials

Compared to Ch, ChS NPs have been extensively studied for their varied applications in medicine and pharmaceutics. Although the reason why exactly ChS is preferred over Ch is not clearly laid out, there is definitely a clear edge that ChS holds over Ch when it comes to publications reporting medical applications. The insoluble feature of Ch in common solvents greatly restricts its application into biomedical of different fields [50], this being the case, two other derivatives of chitin (carboxymethyl chitin (CMC) and chito oligosaccharide (COS)), are those that are reported with respect to biomedical applications. ChS being biocompatible, allow encapsulation and chain grafting of drugs and active ingredients. In addition, ChS NPs possess interesting medical applications. ChS NPs are known to reduce enzymatic degradation of drugs [51], as well as reduce the damage of non-targeted tissues [2]. This comes very handy when it comes to their usefulness for drug delivery, cancer treatment, and biological imaging and diagnosis [52,53]. The highly positive surface charges of ChS NP surfaces, aid in carrying substances into the human system [54]. Studies on nanoencapsulation with ChS NPs are mostly related to the delivery of therapeutic peptides such as insulin [55,56,57] and Cyclosporin A [58,59], and DNA [60,61,62] for gene therapy.

ChS NPs have been demonstrated as carriers for the controlled delivery of doxorubicin (DOX), an anticancer drug used for the treatment of several tumors. DOX treat several cancers such as lymphomas, acute leukemia, osteogenic sarcomas, pediatric malignancies, breast and lung carcinomas. The challenge here is that only small amounts of DOX reach the tumor target site. This is because ~40% is excreted via liver metabolism. In addition to this, DOX also induces cardiac toxicity [63,64,65]. ChS NPs conjugated with DOX have been proved to overcome all these above-said limitations [66,67]. ChS NPs have also been used for drug delivery of 5-fluorouracil (5-FU) and leucovorin (LV), both of these that are used for the treatment of colon cancer [63], as well as for delivery of avidin and biotin, which are used for hepatic carcinoma treatment [68]. ChS NPs have also been demonstrated for their gene delivery applications [69], small interfering RNA (siRNA) delivery [70], vitamin C release [71], plasmid DNA (pDNA) delivery against hepatitis B [72], and insulin [73] or bovine serum albumin (BSA) delivery [74,75] delivery of polyphenolic antioxidants called catechins to the intestine [76]. Other researchers showed that ChS NPs loaded with insulin improved the systemic delivery of insulin through the nasal passage [73]. To protect metronidazole (MZ) from dissolution in saliva, the drug was loaded into ChS NPs for controlled release of the drug over 12 h, to dampen their side effects [77,78].

As a therapeutic delivery entity, ChS NPs attract great attention due to their approach via different pathways of intake such as oral, nasal, and intravenous [79]. The positive charge of ChS NPs, gives them the advantage of high affinity for negatively charged cell membranes [80]. The hydrophobic nature of ChS NPs influences the efficient encapsulation of hydrophilic therapeutics [81]. ChS NPs can increase drug permeability across absorptive epithelia [82,83]. Since ChS NPs are internalized via endocytosis, they are able to deliver biologically active materials into cells without compromising the integrity of the cargo or the cell [84]. Yu et al. synthesized a copolymer of poly (l-lysine) with ChS and studied its efficiency in relation to plasmid DNA adherence capability and gene transfection effect [85]. Rizeq et al. [86] and Joseph et al. [6] have extensively reviewed the various biomedical applications of ChS NPs.

## 3. Chitin/Chitosan NFs

### 3.1. Overview of Ch/ChS NFs Fabrication Methods

The usual techniques that are validated for producing NFs include: electrospinning, printing, self-assembly, phase separation, and template synthesis. With respect to Ch/ChS NFs fabrication: electrospinning, self-assembly, and phase separation are the more predominantly used methods. Few other add-on techniques have also been applied in the case of Ch/ChS NFs. These include microcontact printing, simple mechanical treatment, ultrasonication, and 2,2,6,6-tetramethylpiperidinooxy (TEMPO) mediated oxidization. Electrospinning, uses electrostatic forces to generate polymer NFs, and is an age-old technique, with an experience of 60 years [87,88]. In case of Ch/ChS NFs, electrospinning has been the most frequently used and published method. The electro-spinning process is the typical method of artificially producing NFs from a polymer solution [89]. If the characteristic NF structure of chitin is maintained after deacetylation, a downsizing process may be useful for the production of chitosan NFs. Dry chitosan powder was treated using the high-pressure waterjet system and disintegrated into NFs. An electrospun nanofibrous mat of pure chitosan was successfully prepared by Ohkawa and coworkers [90] using trifluoroacetic acid (TFA) as electrospinning solvent. Under optimized conditions, homogenous chitosan fibers with a mean diameter of 330 nm were prepared. Sangsanoh et al. [91] developed electrospun chitosan nanofibers (average diameters of 126 ± 20 nm) using TFA/DCM (70:30 *v*/*v*) as electrospinning solvent. Haider and Park [92] prepared similar chitosan nanofiber mats by electrospinning. Vrieze et al. [93] attempted a range of acid aqueous solutions for developing chitosan nanofibers by electrospinning.

Molecular self-assembly is another powerful approach, mediated by notable hydrogen bonds, weak non-covalent bonds, ionic bonds, van der Waals interactions and hydrophobic interactions [94,95]. Ch NFs with diameters of 3 nm have been fabricated in hexafluoroisopropanol (HFIP) using facile self-assembly [96,97]. Phase separation technology works on the thermodynamic demixing of a homogeneous polymer-solvent solution into a polymer-rich phase and polymer-poor phase [98]; this has been used to prepare ChS NFs [99,100]. Microcontact printing is a powerful technique successfully applied to the fabrication of nanostructured dendrimers, peptides, and conducting polymers [101,102,103]. Ch NFs have been obtained using the microcontact printing method by dissolving Ch in HFIP [104]. Ultrasonication has also been widely used to obtain individualized Ch/ChS NFs [104]. Using this simple technique, Ch NFs that are 3–4 nm wide and few microns in length were prepared [105]. The TEMPO-mediated oxidation was originally devised for cellulose NFs, Fan et al. successfully prepared Ch NFs by TEMPO-mediated oxidation [106].

Ifuku and his research group have elaborately worked on the fabrication of Ch NFs from various sources using a simple mechanical technique of grinding. They have successfully demonstrated the preparation of Ch NFs from crab shells by a disintegration process [107]. Crab shells have a hierarchical organization with various structural levels. Figure 2 displays the representatives of chitin organization in various chitin sources. They (Ch NFs) are enveloped within protein sheaths and occur as such in the exoskeletons of crabs.

The shells were first purified by a series of conventional chemical treatments and then subjected to mechanical treatment. Proteins and minerals were removed using NaOH and HCl treatments [108,109] and then a grinder was used [110,111]. A pair of grinding stones disintegrated the hierarchical organization of Ch in the crab shells. Highly uniform Ch NFs with a width of approximately 10 nm were obtained. This simple but powerful method obtains homogeneous chitin NFs from waste crab shells in large quantities. The same group also successfully prepared Ch NFs from prawn shells [112] using a method similar to that of the crab shells, since prawn shells are also made up of a hierarchical organization. *Penaeus monodon* (black tiger prawn), *Marsupenaeus japonicas* (Japanese tiger prawn), and *Pandalus eous Makarov* (Alaskan pink shrimp) shells were demonstrated for extraction of Ch NFs. These species serve as seafood delicacies and their shells are thrown away as food waste [113,114]. These authors also reported the preparation of Ch NFs from mushroom [115]. The cell walls of mushrooms consist of Ch NFs that occur complexed with glucans [116,117]. Five widely consumed species of mushrooms: *Pleuotus eryngii* (king trumpet mushroom), *Agaricus bisporus* (common mushroom), *Lentinula edodes* (shiitake), *Grifola frondosa* (maitake), and *Hypsizygus marmoreus* (bunashimeji) were used to extract Ch NFs. Because the organization of chitin in the mushroom cell walls is different from those of crab and prawn shells (which have a hierarchical organization of chitin), the extraction method was modified [118]. Ch NFs were also extracted from squid pens [119,120]. β-type chitin is present in squid pens. Fan and Isogai et al. prepared Ch NFs from squid pen β-chitin [119]. Mechanical treatments under acidic conditions lead to extraction of Ch NFs. Cationic charges accelerate NF conversion and lower crystallinity, parallel chain-packing mode and relatively weak intermolecular forces of squid pen β-chitin play a crucial role in the preparation of chitin NFs. Ch NFs have also been prepared from commercial chitin [121,122,123]. Dry chitin was dispersed in acidic water and passed through a grinder. Acquiring NFs from commercial chitin is advantageous because a large amount of chitin can be obtained within a few hours. A high-pressure waterjet system called the Star Burst instrument was used successfully for the nanofibrillation of dry chitin. Ifuku and group have extensively worked on and published on Ch NFs and their preparations and modifications [107,124]. Figure 3 gives the overall work flow towards procuring Ch/ChS NFs.

### 3.2. Medical Milestones of Ch NFs

When it comes to nanofibers, Ch NFs and their preparation methods have been well documented. However, the medical applications of Ch NFs include scattered reports supported by few publications. The available scanty information are summarized below. Ch possesses nonspecific antiviral and antitumor activities [125,126] and nanosized chitin influences its effects on immune cells [127,128]. Ch NFs have been proposed for tissue engineering scaffolds, drug delivery and wound dressing application [129]. There are a few reports on the successful use of Ch/ChS NFs in the area of tissue engineering because the structure is similar to glycosaminoglycans [130,131]. Ch/ChS NFs attach and proliferate osteoblast cells, support the development of mineralized bone matrix. They also are easy to mold into various geometries and are biocompatible, biodegradable, and exhibit strong antibacterial activity [131,132,133,134]. Bone tissue engineering procedures usually combine hydroxyapatite (HA) with the chitin and chitosan scaffolds to improve the activity and viability of cells, as well as enhance mechanical and cell attachment properties of scaffolds [135,136].

The in vivo effects of Ch NFs after oral administration are undocumented. The preventive effects of Ch NFs in a mouse model of DSS-induced acute ulcerative colitis (UC) was studied [137]. They lead to decreased colon inflammation and histological tissue injury in mouse models. Ch NFs have anti-inflammatory properties and are able to suppress NF-κB and MCP-1 activation and fibrosis in acute UC mouse models. This confirms that ChNF can be prescribed as a potentially novel medicine or functional food for patients with inflammatory bowel disease [138,139]. Furthermore, the application of Ch NFs to skin improved the epithelial granular layer, increased granular density and resulted in lower production of TGF-β. This implies that Ch NFs can be incorporated into cosmetics or textiles [140,141]. Ch NFs are reported to possess other powerful biological activities and few applications have been proposed with respect to biomedical applications [142]. However, as of now, their (Ch NFs) biomedical accomplishments still are restricted to limited research validations.

### 3.3. Medical Milestones of ChS NFs

ChS NF are reported for their excellent biological properties related to biocompatibility, biodegradability, cellular binding capability, wound healing, antitumor, and antibacterial applications. ChS NFs incorporated with hydroxyapatite have been used in bone tissue engineering because of their structural similarity to the extracellular matrix of the native bone tissue [143,144]. ChS NFs containing hydroxyapatite were electrospun to test its ability to regenerate bones [145]. Hydroxyapatite blended ChS NFs crosslinked with genipin facilitated the formation of bones [146]. ChS NFs were electrospun and used as scaffolds in vascular tissue engineering [147]. A 3D gradient heparinized chitosan/poly-3-caprolactone NF scaffold was used [148]. Cooper et al. have fabricated randomly oriented and aligned chitosan/poly-3-caprolactone (chitosan/PCL) fiber scaffolds and investigated their potential use in nerve regeneration [149]. Chitosan NFs with diameters of 4 nm and 12 nm were coupled with poly-D-lysine (PDL) and employed in mouse cortical neuron cultures to examine their capability to support cell attachment, neurite coverage and survival, suggesting significant improvement in long-term cell viability. PLGA/Ch NFs were tested for their potential as drug release systems [150]. ChS blended with ethylenediaminetetraacetic acid (EDTA) and polyvinyl alcohol (PVA) (ChS–EDTA/PVA) as NF scaffolds have been reported [151]. *Garcinia mangostana* fruit hull extracts and lysozyme were incorporated into ChS–EDTA/PVA NF [152] and used as wound dressings. Chitosan is endowed with natural antimicrobial activity [153], Ch NF antibacterial activity has come in handy for wound dressing, antibacterial filtration and tissue engineering applications [154]. ChS NF biosensors have also been developed [155,156]. ChS/PVA NF were electrospun for lipase immobilization [157]. A biosensor based on ChS NFs incorporated with cholesterol oxidase and Au NPs was developed to detect cholesterol [156]. The clinical use of chitin and chitosan-based NFs still remains challenging and needs further insights. Not many supportive clinical trials have been achieved. Kossovich et al. developed ChS/PEO NFs towards wound dressings for IIIa and IIIb degree burns [158]. The results were highly promising. However, nothing much has been progressing in this direction of putting ChS NFs to clinical use. Bazhaban and her colleagues in 2013 fabricated a biodegradable nanofibrous controlled release drug delivery system using chitosan and Beta cyclodextrin (β-CD); salicylic acid was the model drug. The results were promising, and suggested that the electrospun nanofibers of PVA/ChS NFs-g-β-CD could be of high potential for biomedical applications [159]. Its almost a decade this report has come out and yet, once again no progress has been made in this direction. Nawzat et al., 2019, have presented an overview of ChS NFs and their drug delivery applications [160].

## 4. Ch/ChS NF Modifications and Nanocomposites

Chitin nanomaterials have a reactive surface covered with hydroxyl groups, opening up the possibility for extensive chemical modification. Using surfactants or by chemical grafting/modifications, the mechanical performance of the Ch nanocomposites can be enhanced. Reinforced Ch/ChS based nanocomposites blended with poly (methyl methacrylate), epoxy, polystyrene, polyaniline, polysulfone, polycarbonate, and thermoplastic polyurethane [161] have been reported. Apart from these, Ch/ChS composites with hydrophilic and bio-based polymeric matrices, namely, chitosan, starch, PLA, and cellulose, have also been fabricated [49,162,163,164,165,166]. Several chemical modifications of chitin NF surfaces have been achieved: acetylation [43], deacetylation [49], phthaloylation [167], naphthaloylation [89], maleylation [89], chlorination [115], TEMPO-mediated oxidation [168], and graft polymerization [169]. Modification is a promising and effective way to design functional materials and enhance their functionality. The blending in of two to three different materials, allows the harnessing of their individual properties in one material. ChS nanofibers have been developed using chitosan and synthetic polymers such as poly(vinyl alcohol) (PVA), poly(ethylene oxide) (PEO), poly(ethylene terpthalate) (PET), polycaprolactone (PCL), poly(lactic acid) (PLA), and nylon-6. These composite nanofibers are more advantageous than pure chitosan, because their mechanical, biocompatible, and antibacterial properties were drastically enhanced by the addition of these polymers.

Ding et al., 2014 have published a comprehensive summary of the various Ch/ChS NF materials and their medical applications [170]. Chitin based binary blend NFs were fabricated using HFIP as solvent. Park et al., reported Ch/PGA blend NFs using HFIP as the solvent via electrospinning method [171]. Using the same solvent, Ch/silk blend NFs were fabricated as well [172]. Synthetic polymers, such as PVA, PLGA, PEO, and PGA have been widely used with ChS NFs [115,171,173,174]. These are expected to deliver much towards medical applications. Biopolymers such as silk, cellulose, collagen, hyaluronic acid, alginate, and zein have been electrospun with ChS [175,176,177,178]. Native cellulose/chitosan NF composites have also been successfully produced [179]. Hydroxyapatite/ChS NF scaffolds have been prepared for tissue engineering [145,180].

A further development in chitin/chitosan based nanocomposites, is the production of ternary composite NFs blended with synthetic polymers, biopolymers and inorganic substances [181,182,183,184,185,186]. These composite scaffolds are expected to show better mechanical properties and biocompatibility. Other chitosan/synthetic polymer/natural polymer ternary blend NFs include, carboxymethyl chitosan/poly(vinyl alcohol)/silk fibroin [187], chitosan/lysozyme/poly(vinyl alcohol) [188], poly (lactic acid)/chitosan/collagen [189], and collagen/chitosan/hermoplastic polyurethane [190]. Some inorganic substances such as silica, Au nanoparticles, Ag nanoparticles, multiwalled carbon nanotubes, and hydroxyapatite were incorporated into chitosan/synthetic polymer (PVA and PEO) NFs to enhance the mechanical properties and confer antibacterial activity, protein adsorption ability, cell attachment and proliferation [125,126,127,128,129].

Carboxymethyl chitosan is one of the most frequently used chitosan derivatives to obtain NFs using the electrospinning method. In order to enhance the spinning capacity, some synthetic polymers such as PVA and PEO were blended with carboxymethyl chitosan to form NFs [187,191,192], their medical applicability has also been demonstrated. Quaternized chitosan was another chitosan derivative which was chosen to form NFs owing to its excellent antibacterial activity and biocompatibility [193,194,195,196]. Figure 4 summarizes the Ch/ChS NF based nanocomposites that are currently established.

## 5. Future Outlook and Conclusions

The current status with respect to Ch/ChS NF synthesis, methods, medical deliverables, and consolidation of available nanocomposites for future endeavors have been comprehensively presented in this review. During the process of reviewing the existing scenario with respect to Ch/ChS NFs, staggering ends and grey areas and lacunae have been identified. For instance, there is almost no information or research findings on Ch based NPs, when ChS based NPs have been extensively prepared and applied towards biomedical applications. The one straightforward existing report [197] confirms the preparation and usage of Ch NPs for wound healing applications. This being true, then there could definitely be more to harness from Ch NPs. This is somewhere research needs to be focused, and if there is a challenge, it needs to be exposed and addressed. This review interestingly also found that there was a strong bias in the usage of Ch and ChS NFs for biomedical applications. Ch NFs are very less represented when it comes to medical applications. The application aspect of Ch NFs is significantly lagging. There are few reasons, that of course indicate that chitosan is better than chitin, owing to its solubility and chitosan’s degree of deacetylation which is proportional to its biological activity. However, there is no information on the continuing retardation in the bio applicability of Ch NFs versus ChS NFs. Chitin as such, facing inadequacy is understandable, but with the introduction of nano inputs and preparation of advanced NFs, there is always plenty of room for reversal of the inherent material limitations. This being so, this review provokes researchers to apply Ch NFs for biomedical aspects where previously chitin materials have failed. Drug delivery is another aspect that ChS NFs have been luxuriously used and Ch NFs have been seldom used. This is another research direction.

ChS-based composites, binary and ternary composites have been well documented, but their medical applications remain restricted to scattered scanty publications; this is one area that needs development. Ch based nanocomposites are once again relatively fewer compared to ChS NF nanocomposites; this is another area that can be built on. Ch NF nanocomposites could be a real launching factor, breaking limitations and bringing Ch based biomedical applications to the forefront. Meanwhile ChS NFs also need expansion laterally blending with composites. Synergistic composites combining multiple polymeric and inorganic metal nanoparticle systems, could extend their (ChS NFs) utility for medical applications further. This review expects to draw the attention of workers to expand on these grey areas.

## Figures and Tables

**Figure 1 polymers-13-02330-f001:**
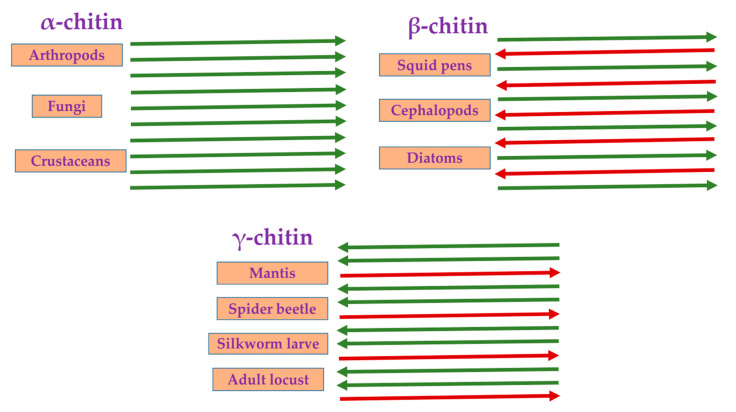
Schematic representation of the three polymorphs of chitin and their respective source organisms.

**Figure 2 polymers-13-02330-f002:**
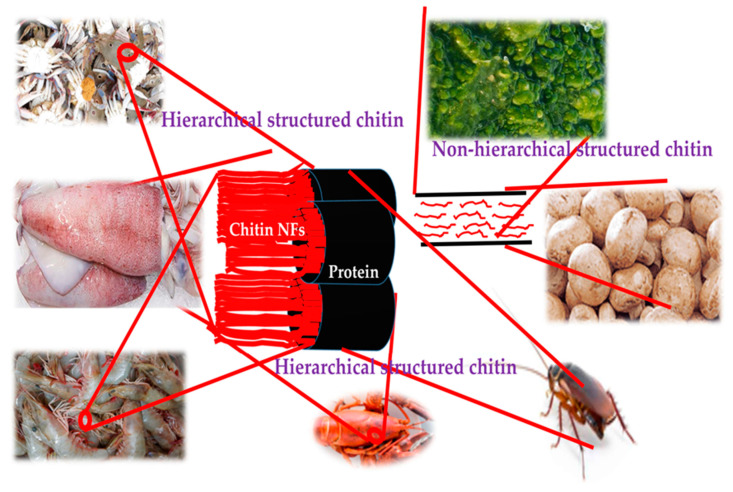
Representatives of natural NFs in chitin sources.

**Figure 3 polymers-13-02330-f003:**
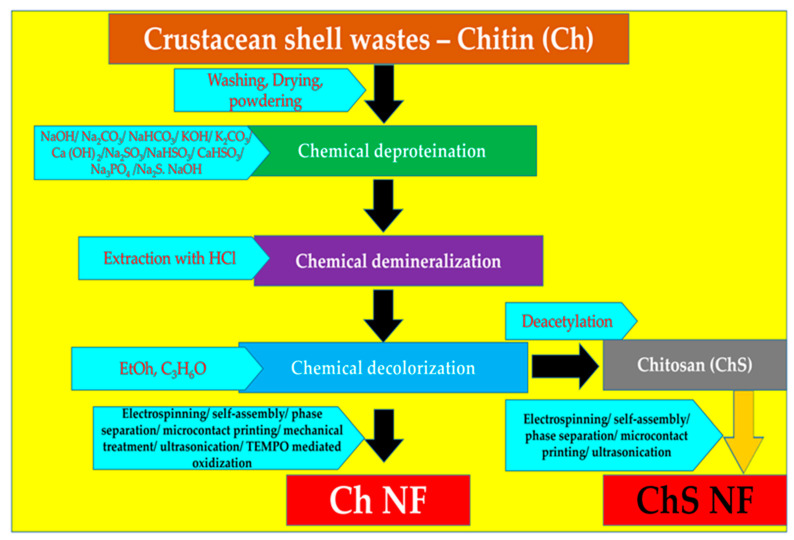
Overview of the overall steps involved in the fabrication of Ch/ChS NFs.

**Figure 4 polymers-13-02330-f004:**
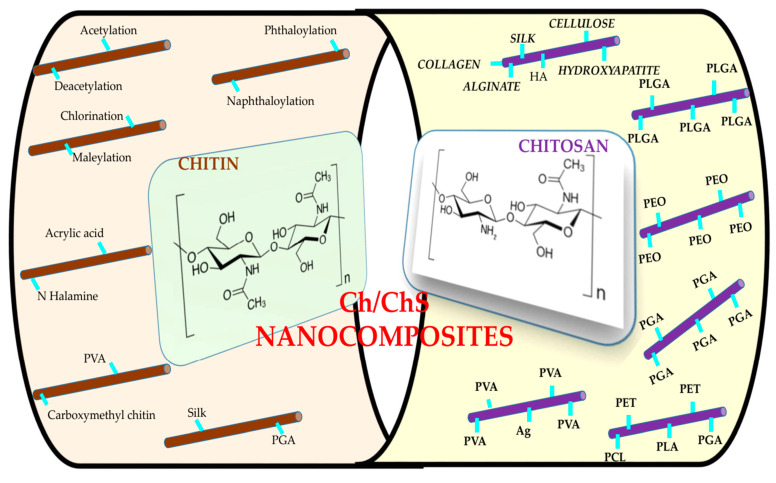
Snapshot of the existing Ch/ChS based nanocomposites.

## Data Availability

Not applicable.

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
