# Peer review of "Reviewing Chitin/Chitosan Nanofibers and Associated Nanocomposites and Their Attained Medical Milestones"

_polymers, 2021, doi:10.3390/polym13142330_

Round 1
Reviewer 1 Report
Major:
- Page 9, line 345, “Ding et al., 2014 have published a comprehensive summary of the various Ch/ChS NF 345 based composites, in their excellent review”. What is the difference between this manuscript and Ding et al.’s work?
- The authors should cite the most recently published manuscript;
- Are all the figures created and captured by the authors? If not, then appropriate citations are required;
- Page 1, line 41 and Page 2, line 42, “there are different orientations of the microfibrils: α-chitin has antiparallel chains, β-chitin has parallel chains”. It can be found from Figure 1 that α-chitin is parallel, while β-chitin is anti-parallel. Please explain;
- Section 2.1, the author shall compare the advantage and disadvantage of each method for Ch/ChS nanomaterial preparation;
- Section 2.2, biomedical applications of Ch was not clearly stated.
Minor:
- Leave the same blank spaces at the beginning of each paragraph;
- page 5, line 218, there shall be a space between Ch and NFs.
Author Response
We would like to thank the Editor and the reviewers for their valuable inputs and suggestions and for the revision opportunity. We have now revised the manuscript as per your suggestions. We present a point by point response to your queries below.
Major:
- Page 9, line 345, “Ding et al., 2014 have published a comprehensive summary of the various Ch/ChS NF 345 based composites, in their excellent review”. What is the difference between this manuscript and Ding et al.’s work?
Ans. Ding et al., have a review on ‘Emerging chitin and chitosan nanofibrous materials for biomedical applications’, in it they have addressed a topic on composites. Our review, focused more elaborately on the nanocomposite part and also reviewed all the recent work in the field. Ding et al., review is dated 2014. I feel the way we have described Ding et al in the text is incorrect. We have modified it now. Thank you.
- The authors should cite the most recently published manuscript;
Ans. We have cited literature upto 2021. We have added a few too. Thank you.
- Are all the figures created and captured by the authors? If not, then appropriate citations are required;
Ans. We confirm that all the figures are original. Thank you.
- Page 1, line 41 and Page 2, line 42, “there are different orientations of the microfibrils: α-chitin has antiparallel chains, β-chitin has parallel chains”. It can be found from Figure 1 that α-chitin is parallel, while β-chitin is anti-parallel. Please explain;
Ans. Sorry about that, the description has been swapped, very sorry. Thank you for rightly pointing out. We have revised the writing part.
- Section 2.1, the author shall compare the advantage and disadvantage of each method for Ch/ChS nanomaterial preparation;
Ans. This is not exactly the objective of this review, since we are focusing on the NFs. Thank you for your kind understanding.
- Section 2.2, biomedical applications of Ch was not clearly stated.
Ans. Yes we understand, in fact there Ch is not actually used for biomedical applications. The insoluble feature of Chitin in common solvents greatly restricts in the applications of biomedical fields, Ch nanofibers are the ones that have some biomedical inputs and that has been dealt with in section 3.2. We have sort of addressed this issue with an added reference in section 2.2. so that the readers have clarity. Thank you.
Minor:
- Leave the same blank spaces at the beginning of each paragraph;
Ans. Corrected.
- page 5, line 218, there shall be a space between Ch and NFs.
Ans. Corrected.
Reviewer 2 Report
In this paper, the authors reviewed the fabrication and applications of chitin and chitosan nanofibers in the biomedical field. Although the topic is interesting and suitable for the journal, some improvements are needed before it is published.
There are various punctuation errors (as the comma between the subject and the verb) and some repetitions. For example:
- Line 13: remove the comma between review and gives;
- Lines 15 and 17: “have been reviewed” is used twice in few lines;
- Line 31: remove the comma between today and has expanded;
- Line 32-33: the sentence “as of now…technology” is meaningless;
- Line 69: Properties of NFs and not of NF;
- Line 90: remove the comma between 2015 and have;
- Line 93: remove the full stop between medication and the reference;
- Lines 97-98: the sentence “the reversed…NPs” is meaningless;
- Line 133: remove the comma between al and reported;
- Line 136: “in width” and not “width”;
- Line 137 and line 184: the word “authors” can be removed;
- Line 231: “treatments” and not “treatment”.
In line 36, some papers related to the used of polysaccharide polymers in pharmaceutical applications have been cited. I think the techniques based on the use of supercritical fluids deserve a mention (see, for example, 10.1016/j.jcou.2019.11.001; 10.3390/polym13111882).
In line 48, I think that 1010-1011 is a too small range. “around 1010” would be better.
In line 69, “more than 100” is not clear.
Lines 223-224: add a ref.
Author Response
In this paper, the authors reviewed the fabrication and applications of chitin and chitosan nanofibers in the biomedical field. Although the topic is interesting and suitable for the journal, some improvements are needed before it is published.
Ans. We would like to thank the Editor and the reviewers for their valuable inputs and suggestions and for the revision opportunity. We have now revised the manuscript as per your suggestions. We present a point by point response to your queries below.
There are various punctuation errors (as the comma between the subject and the verb) and some repetitions. For example:
Ans. Sorry about that, we have corrected that.
Line 13: remove the comma between review and gives;
Ans. Corrected
Lines 15 and 17: “have been reviewed” is used twice in few lines;
Ans. revised
Line 31: remove the comma between today and has expanded;
Ans. removed
Line 32-33: the sentence “as of now…technology” is meaningless;
Ans. modified
Line 69: Properties of NFs and not of NF;
Ans. changed
Line 90: remove the comma between 2015 and have;
Ans. changed
Line 93: remove the full stop between medication and the reference;
Ans. changed
Lines 97-98: the sentence “the reversed…NPs” is meaningless;
Ans. revised
Line 133: remove the comma between al and reported;
Ans. removed
Line 136: “in width” and not “width”;
Ans. Changed.
Line 137 and line 184: the word “authors” can be removed;
Ans. Removed.
Line 231: “treatments” and not “treatment”.
Ans. Changed.
In line 36, some papers related to the used of polysaccharide polymers in pharmaceutical applications have been cited. I think the techniques based on the use of supercritical fluids deserve a mention (see, for example, 10.1016/j.jcou.2019.11.001; 10.3390/polym13111882).
Ans. We have added this citation - Pantić, M.; Kravanja, K.A.; Knez, Ž.; Novak, Z. Influence of the Impregnation Technique on the Release of Esomeprazole from Various Bioaerogels. Polymers 2021, 13, 1882.
In line 48, I think that 1010-1011 is a too small range. “around 1010” would be better.
Ans. Changed.
In line 69, “more than 100” is not clear.
Ans. Clarified.
Lines 223-224: add a ref.
Ans. Added.
Round 2
Reviewer 1 Report
The reviewer has reviewed the revised manuscript and confirmed that certain requested actions have been done.
Reviewer 2 Report
The paper has been improved.
It can be accepted